# Additive Manufacturing Using Agriculturally Derived Biowastes: A Systematic Literature Review

**DOI:** 10.3390/bioengineering10070845

**Published:** 2023-07-17

**Authors:** Al Mazedur Rahman, Taieba Tuba Rahman, Zhijian Pei, Chukwuzubelu Okenwa Ufodike, Jaesung Lee, Alaa Elwany

**Affiliations:** 1Department of Industrial & Systems Engineering, Texas A&M University, College Station, TX 77843, USAzjpei@tamu.edu (Z.P.);; 2Department of Engineering Technology and Industrial Distribution, Texas A&M University, College Station, TX 77843, USA; 3J. Mike Walker ’66 Department of Mechanical Engineering, Texas A&M University, College Station, TX 77843, USA

**Keywords:** additive manufacturing, agricultural wastes, biomass, biowastes, FDM, LDM, stereolithography, selective laser sintering, binder jetting

## Abstract

Agriculturally derived biowastes can be transformed into a diverse range of materials, including powders, fibers, and filaments, which can be used in additive manufacturing methods. This review study reports a study that analyzes the existing literature on the development of novel materials from agriculturally derived biowastes for additive manufacturing methods. A review was conducted of 57 selected publications since 2016 covering various agriculturally derived biowastes, different additive manufacturing methods, and potential large-scale applications of additive manufacturing using these materials. Wood, fish, and algal cultivation wastes were also included in the broader category of agriculturally derived biowastes. Further research and development are required to optimize the use of agriculturally derived biowastes for additive manufacturing, particularly with regard to material innovation, improving print quality and mechanical properties, as well as exploring large-scale industrial applications.

## 1. Introduction

Additive manufacturing (AM) can produce products from 3D model data by applying a layer-by-layer approach, distinct from formative and subtractive manufacturing methods [1]. The main advantages of AM include producing products with complex geometries from different materials (such as polymers, ceramics, and metals), energy efficiency, product personalization, small-scale productions, and the potential to implement a distributed network of affordable equipment in local communities [2]. The importance of AM in fostering a more sustainable economic system is becoming more apparent in light of its numerous advantages and its expanding applications [3]. AM has been recently used for valorization of various resources at their end of life (EoL), such as recycled plastics [4], biomaterials for construction sectors [5], and electronic wastes [6]. AM promotes the use of agriculturally derived wastes. Additionally, AM can reduce pollution brought on by wastes (biomass) created during traditional industrial transformation of agricultural products.

According to FAOSTAT [7], currently, the agricultural industry worldwide utilizes almost one third of the earth’s land surface and generates a significant amount of biowastes such as rice husk/bran, millet stoves, sugarcane bagasse/tops/molasses [8], wheat bran/straw, and oat straw [9]. When handling agricultural biowastes, it is important to consider that they contain substantial quantities of carbon (C) and plant nutrients. These biowastes are precursors to greenhouse gases (GHGs), ammonia (NH3), pollution of surface water, offensive odor, and particles [10,11,12,13]. The management of agricultural biowastes has a significant impact on the extent of the resulting emissions. When devising sustainable methods for recycling and utilizing agricultural biowastes, it is important to consider the risk of disease transmission [14], heavy metal, and biogenic contamination of the soil [15], as well as the pollution caused by excessive nutrient use and gaseous emissions.

Combining AM with biowastes demonstrates the potential to encourage the application of eco-friendly design, such as designing of products by upcycling or recycling biowastes [16,17]. FDM (Fused Deposition Modeling), also known as Fused Filament Fabrication (FFF) [18], Direct Ink Writing (DIW), also known as liquid deposition modeling (LDM) [19,20], stereolithography [21], and binder jetting AM [22] methods are used for 3D printing of agriculturally derived biowastes (Figure 1).

A recently published paper contained a thorough evaluation of the literature on biomass materials—either wastes or byproducts—for FDM and LDM [2]. However, this evaluation did not cover other additive manufacturing methods and was not exclusively focused on the newly developed novel biowaste materials. Another review paper discussed biopolymeric sustainable materials and their emerging applications [25]. However, its focus was not on agriculturally derived biowastes. It mentioned additive manufacturing but did not exclusively focus on additive manufacturing.

There are several reasons to write a literature review that places particular emphasis on different additive manufacturing methods that utilize agriculturally derived biowastes. Firstly, it promotes sustainability by utilizing agricultural biowastes, reducing reliance on traditional petrochemical-based materials and minimizing environmental impact. Secondly, it facilitates the development of novel materials with unique properties, fostering innovation in product design and expanding the range of applications. Additionally, the incorporation of biowastes in large-scale industrial applications brings economic advantages, contributing to the establishment of a bio-circular economy and promoting a more sustainable and efficient approach to manufacturing.

Therefore, this systematic review includes different AM methods used for printing agriculturally derived biowastes, novel materials developed for printing biowastes, important printing parameters and printing technology considered for printing these biowastes, characterizations required for evaluating the mechanical properties of these biowastes, potential applications and limitations of incorporating these biowastes into large-scale applications and future research opportunities focused on improving print quality and enhancing mechanical/thermal properties for large-scale production integrating biowastes.

This study reports a thorough evaluation of the literature on novel materials made from agriculturally derived biowastes—either wastes or byproducts—for various AM methods. The main research questions (RQ) are as following:

RQ1: What is the state-of-the-art research regarding AM of agriculturally derived biowastes?

RQ2: What are the types of agriculturally derived biowastes, novel 3D printable materials incorporating biowastes, commonly used AM methods, and important printing parameters?

RQ3: What are the potential applications, as well as constraints, in the context of using agriculturally derived biowastes for AM methods in large-scale production?

RQ4: What are the common mechanical characterization tests performed on these biowaste materials?

RQ5: What are future research opportunities?

A total of 57 papers are included in this study after screening, and the screening process is described in the methodology. All these papers are presented in a table to show the list of recent research studies (RQ1). These papers are analyzed based on the agriculturally derived biowastes, novel 3D printable materials, and types of AM methods (RQ2). Beginning with the possible applications of these materials, various limitations in the context of using AM methods in large-scale applications integrating these biowastes are considered (RQ3). Then, common mechanical characterization tests are analyzed for these biowastes (RQ4), and lastly, future research opportunities are described (RQ5). This review aims to provide an overview of agriculturally derived biowastes and their potential applications in AM for large-scale applications.

## 2. Methodology

This study represents a systematic literature review in accordance with the PRISMA systematic review statement [26]. PRISMA 2020 implementation could be advantageous for authors, editors, and peer reviewers of systematic reviews. Readers can evaluate the applicability of the methodologies and, consequently, the veracity of the conclusions. The goal of this PRISMA statement is to increase the completeness, accuracy, and transparency of systematic reviews. PRISMA primarily comprises a checklist and a flow diagram that depict the workflow for the search, identification, screening, and analysis procedures and aid in determining whether the information is exhaustive [27].

Table 1 summarizes the criteria of eligibility, search library, and binary strings (selected based on [28]) for query used to select papers that are included in this study. This study does not include review and meta-analyses papers. All the papers included have an experimental component connected to different types of AM methods. A preliminary screening revealed no papers published prior to 2016. Hence, a timeframe of 2016 to 2023 was decided upon. Since the study explores agriculturally derived biowastes in relation to the AM methods, papers related to wastes from chemicals or other industrial systems were excluded. Papers on biowastes obtained from wood (forestry industry), fish, and algal cultivation were included for this review. All these biowastes are mentioned as agriculturally derived biowastes throughout the review. Lastly, the papers lacking a description of the AM methods, such as the type of printer, type of materials, or newly developed materials, were excluded. The search was conducted using TAMU (Texas A&M University College Station) library and Web of Science (WoS) library.

First, duplicates and inaccessible works were removed from search results. In order to choose the records that would be included and removed, the screening process began with a check of the titles, abstracts, and keywords. The papers selected from the first screening were then taken into a full text review based on agriculturally derived biowastes, AM methods, and novel materials incorporating biowastes. By thoroughly reading the full texts, two further screenings were conducted. Then, at the data extraction stage, data columns of paper categories, links, keywords, material specifications, suppliers, sources, AM methods, applications and limitations, and future research opportunities were created (Figure 2). These data were extracted using full texts, as well as some websites and source sites. On some occasions, the data for comparison with the other articles such as printing parameters were normalized based on units.

## 3. Results and Discussion

The screening described in the previous section resulted in 57 papers. The analysis of these 57 papers is presented in this section.

### 3.1. Initial Analysis

Figure 3a and Table 2 show that the number of research papers on 3D printing using agriculturally derived biowastes became higher after 2019. There were 23 articles in year 2022, which equals to the number of papers in year 2021 and 2020 combined.

These papers were classified according to research areas (Figure 3b). To evaluate this aspect, the following factors were considered in detail: the journal’s research field, the specific papers’ keywords, and the research contents. For visual purposes, the principal research areas that had only one publication were mentioned as “Others” in Figure 3b. There were four main research areas: food engineering, materials science and engineering, polymer science and engineering, and composite science and engineering. About 81% of papers were from these four areas. If three research areas (polymer science and engineering, materials science for specific sectors, and composite science and engineering) are considered as subsets of materials science and engineering, around 58% of selected papers are from the materials research area. 

### 3.2. Agriculturally Derived Biowastes

This subsection provides an overview of the biowastes that have been employed in additive manufacturing, including their types and sources. It also discusses the particle size and shape that are typically utilized in various AM methods, as well as matrix materials and additives employed in formulating printable filaments, powders, or inks. Finally, this subsection shows the maximum weight percentages at which biowastes, used as biofillers, can be incorporated into printable filaments, powders, or inks based on different additive manufacturing methods, as well as advantages and challenges of using biowastes as biofillers.

Table 3 lists the types of materials used in selected papers. Mostly used biowastes are biomass, shell, husk, and fiber (Figure 4a). All the definitions of the biowastes mentioned in Figure 4 are taken from the cited references in Table 3. These biowastes originate from various sources such as wood, rice, nuts, algae, fungi, vegetables, fish, and crabs. 

In this review, the sources of the materials are agricultural industry (wheat, corn, rice, plants, vegetables, banana, nuts, etc.), forestry and furniture industry (wood), and fish industry (crab, cod, salmon, etc.). Figure 4b shows that around 54% of the materials in the selected papers are from the agricultural industry and around 67% are from the agricultural and forestry industry combined. Another important source is the food industry related to agrifood, algal cultivation, and fish. A total of 15% of the materials in the selected papers are from the food industry.

Particle size affects printability of the filaments, powders or inks, and mechanical properties of printed parts [53,81], printing resolution, surface roughness of the printed parts, material homogeneity, and nozzle clogging and blockage [38,45,52,53,81,82]. These biowastes were usually ground, sieved, and milled to obtain the required particle sizes [37,42,65,69,72]. For wood powders, particle sizes of 0.6 to 1.25 mm were used in binder jetting printing [74]. For FDM, particle sizes are typically less than 0.65 mm. For instance, buckwheat husk was used for 3D printing filament fabrication, where the particle sizes were mostly 0.2 mm [37]. Different particle sizes (0.02 to 0.65 mm) of flax fiber were used in another paper for a similar FDM method [72]. Mostly, particle sizes ranged between 0.02 and 0.65 mm for FDM methods [29,42,59,67,75,79]. There was one paper that used a 0.009 mm particle size for Miscanthus biomass powder [65]. In most cases, smaller particle sizes were chosen to avoid nozzle clogging during extrusion. 

Particle size used for LDM printing methods can be categorized into two groups: smaller than 0.125 mm and larger or equal to 0.125 mm. For example, banana peel, beechwood sawdust, potato peel, and okara powder used in LDM printing had a particle size less than 0.125 mm [38,45,46,51]. In some selected papers, larger particles (>0.125 mm) of Nostoc sphaeroides, okara, and banana peel were used along with smaller particles [32,51,52].

For selective laser sintering, peanut husk and walnut shell powder were used with particle sizes of less than 0.125 mm [30,60]. For stereolithography, three ranges of particle sizes (<0.045 mm, 0.045–0.125 mm, and >0.125 mm) were used. Additionally, the characteristics of the material depend on the size of the fibers. Larger fiber diameters result in more flaws at the resin-biomass contact in terms of structural characteristics. Larger fibers have demonstrated better performances in relation to the thermal stability of the photo-curable composites due to the higher lignin content [53]. 

One of the selected papers provided evidence that the choice of the matrix is an important parameter to control the mechanical performance of the printed objects [72]. According to Table 3, around 21% of selected papers used poly lactic acid (PLA) as a matrix, which makes it the most popular matrix material for FDM printing. In some selected papers, acrylonitrile butadiene styrene (ABS) [59,70,77], Mater- Bi^®^ EF51L (MB) [42,79], poly(3-hydroxybutyrate-co-hydroxy valerate) (PHBV), hydroxy valerate (HV) [66], poly-(butylene-terephthalate) (PBAT) [76], and recycled polypropylene [3,78] were used. For the LDM printing method, mostly water was used as a matrix for creating printable pastes. In some selected papers, the Na-alginate solution [50], Guar gum (GG), Xanthan gum (XG), wheat flour [44], and glacial acetic acid [83] were used along with water. In selective laser sintering, stereolithography, and binder jetting, various compositions of materials were used, such as epoxy resin, paraffin wax, stearic acid, polyether sulfone, polyamide, methacrylated ethyl cellulose macromonomer (ECM), hexamethylene diisocyanate (HDI), rosin-derived monomers (DAGMA), and 2-hydroxyethyl acrylate (HEA). In a few selected papers, psyllium husk, wheat, water, and fungi were used as matrix [41,84].

Table 3 also shows that a maximum of 30% [34,72] of biowastes were used as biofillers for developing 3D printable filaments for FDM. In all the other FDM-based papers, the percentages were less than 20%. However, a higher amount of biowastes in the matrix may have a negative impact on the mechanical properties of the printed parts. For LDM, the biowaste material content used varied from 0% to 90% for making printable pastes/inks/gels. Most matrix materials are used to change the rheological behavior of the pastes so that printable gels/pastes can be made. Researchers are trying to use various combinations so that printability can be improved [33,49,51,84].

Using biowastes in printable filaments, pastes/gels/inks, and powders can reduce the cost of printed products and also increase their sustainability and biodegradability [51]. However, integrating biowastes into filaments or inks for 3D printing can give rise to certain challenges and difficulties, including altering material properties, potentially affecting the filament’s mechanical and thermal characteristics [30,40,42,48,58,68,69]. This alteration can impact print quality, resulting in reduced strength, increased brittleness, or changes in dimensional stability. High biowaste content can impair interlayer bonding, causing weaker printed parts and delamination. Moreover, complex geometries may suffer from reduced printability, accuracy, and resolution when biowaste content is increased [3,29,42,65,70,72,77].

### 3.3. AM Methods Used for Printing Agriculturally Derived Biowastes

This subsection describes the different AM methods that have used biowastes. It also shows what types of printers are used, the important printing parameters (such as nozzle size, printing speed, layer height, pressure, laser power, scan speed, hatching distance, and temperature) considered for printing materials incorporating biowastes.

Figure 5a shows the percentages of the selected papers using different AM methods. A total of 86% of these selected papers are based on extrusion-based AM methods including FDM and LDM. Only three papers used stereolithography, three papers used selective laser sintering, and two papers used binder jetting (including one paper that used ink jetting, direct cryo writing (DCW) along with binder jetting). About 23% of the selected papers used custom-made printers not available commercially (Figure 5b). 

#### 3.3.1. Extrusion-Based AM Methods

In extrusion-based AM methods, the final product is produced by extruding materials via a nozzle. These methods are under the “Material Extrusion” category of AM as defined by ASTM [1]. FFF (or FDM) and LDM (or DIW) are in this group. Table 4 shows that 49 papers out of the 57 selected papers used extrusion-based AM methods for printing. The printers are small and inexpensive and can print a variety of materials using filaments or pastes.

FDM printers are easily accessible to a wider range of users, ranging from affordable desktop-size devices to massive industrial equipment. Hence, developing novel materials for FDM is appropriate for scaling up in the short- and mid-term, allowing the proliferation of new applications [85]. This is especially true for biowastes from agriculturally derived materials, as 46% of the 49 papers selected for this subsection are based on FDM (Figure 6a).

LDM is another extrusion-based printing method. Figure 6a shows that approximately 46% of the selected papers used the LDM method. LDM can handle a large variety of materials. Processing various material types also implies dealing with a range of potential applications. Compared to papers based on FFF, the papers based on LDM exhibit a more pronounced customization in the 3D printer type. LDM is a more adaptable method for agriculturally derived biowastes with a wider range of applications, including large building structure printers and small bioprinters. Figure 6b shows that around 42% of the selected papers based on the LDM printing method used a customized setup for printing agriculturally derived biowastes. In order to better control the LDM printing parameters for these novel materials, customized systems were developed. In these papers, viscous pastes were printed at room temperature (except one of the selected papers, in which heating during LDM printing was used [64]) by using screw extruders or pump systems on robot arms or customizing existing 3D printers [38,41,58,80,83]. Though a lot of pre-processing or post-processing might be required with LDM, without the need for heating (while printing), LDM use less energy and are more environmentally friendly.

As shown in Table 4, nozzle diameter for FDM printing is typically 0.4 to 1.00 mm. These nozzles are available commercially, and 0.4 mm, 0.6 mm, 0.8 mm, and 1 mm nozzles are the most common ones. However, one of the selected papers used a 1.3 mm nozzle for FDM printing to avoid nozzle clogging because a larger ramie fiber size was used as a filler material [75]. However, in LDM, a larger variety of nozzles is observed. Most of the nozzles used in LDM are larger than 1 mm. Some customized nozzle diameters are even larger than 20 mm [33,68]. Bigger nozzles, along with thicker layers, are common in DIW, making them more suitable for large-scale printing applications such as construction, food, and furniture [86]. Some papers used modified extrusion-based (syringe-based and gear-based LDM) printing methods for specific applications [53,64].

Table 4 also shows a significant difference in printing speed between FDM and LDM-DIW. The printing speed in FDM is usually higher, ranging from 35 to 150 mm/s (mostly 35, 45, 50, and 65 mm/s). Only one paper on FDM used a printing speed range of 1.67 to 8.36 mm/s [75]. On the other hand, the printing speed in all the selected papers that used LDM ranged from 1.67 to 30 mm/s. Only one of the selected papers used a printing speed of 25 to 58.33 mm/s [33].

Table 4 also shows a clear difference in layer height between FDM and LDM. Layer height in FDM normally ranged from 0.1 mm to 1.00 mm, but LDM had a much wider range, up to far more than 1 mm [41]. Another critical parameter in LDM is pressure, with typical values of 2 bar to 6 bar.

**Table 4 bioengineering-10-00845-t004:** List of the selected papers by Extrusion-based AM methods, Printer Size, 3D Printer Name, 3D Printer Type, Nozzle Diameter, Speed, Layer Height, and Pressure.

AM Methods	Printer Size	References	3D Printer Name	3D Printer Type	Nozzle Diameter (mm)	Speed (mm/s)	Layer Height (mm)	Pressure (Bar)
FDM	Small	[39]	//	Extrusion-based printer (standard)	//	//	//	//
		[59]	Leapfrog Creatr 3D	Extrusion-based printer (standard)	0.50	//	//	//
		[72]	Prusa i3 Rework 3D printer	Extrusion-based printer (standard)	1.00	50 to 150	0.6 to 1.0	//
		[29]	//	Extrusion-based printer (standard)	0.60	60.00	0.25	//
		[73]	ENDER-3S	Extrusion-based printer (standard)	0.40	50.00	0.20	//
		[34]	CREALITY CR-10 3D printer	Extrusion-based printer (standard)	0.40	35.00	0.20	//
		[37]	Prusa i3 MK3	Extrusion-based printer (standard)	0.80	50 and infill 80	0.15	//
		[75]	Combot-200 printer	Extrusion-based printer along with dry fiber deposition mechanism	1.30	1.67 to 8.36	0.60	//
		[65]	Lulzbot Taz 6	Extrusion-based printer (standard)	0.50	35.00	0.30	//
		[4]	Lulzbot Taz 6	Extrusion-based printer (standard)	0.50	35.00	0.30	//
		[66]	Ender 5 Pro	Extrusion-based printer (standard)	0.40	65.00	0.25	//
		[76]	Ultimaker 3	Extrusion-based printer (standard)	0.80	//	0.20	//
		[77]	Prusa i3 MK3S	Extrusion-based printer (standard)	0.60	60.00	0.30	//
		[3]	3D FF-STD Doppia machine	Extrusion-based printer (standard)	0.80	60.00	0.25	//
		[78]	3D FF-STD Doppia machine	Extrusion-based printer (standard)	0.80	60.00	0.25	//
		[79]	Sharebot Next Generation	Extrusion-based printer (standard)	0.40	45.00	0.10	//
		[67]	FDM printer	Extrusion-based printer (standard)	//	//	//	//
		[69]	FDM printer	Extrusion-based printer (standard)	//	//	//	//
		[70]	3D printing pen	Extrusion-based printer (standard)	//	//	//	//
		[40]	Ultimaker 3	Extrusion-based printer (standard)	0.60	50.00	0.10	//
		[42]	Sharebot Next Generation	Extrusion-based printer (standard)	//	45.00	0.10	//
		[71]	FS-200 3D printer	Extrusion-based printer (standard)	0.40	50.00	0.10	//
		[82]	Sharebot Next Generation	Extrusion-based printer (standard)	0.40	50.00	0.10	//
LDM	Small	[32]	Model FSE 2	Extrusion-based printer (standard)	2.00	//	3.00	//
		[64]	3D discovery™ Evolution printer	Extrusion-based printer (standard)	0.60	2.50	0.20	4 bars
	Medium	[68]	Custom-built/in-house printer	Extrusion-based printer with 6 nozzles custom built	225 to 400 mm^2^ for 6 different types of nozzles	//	//	//
	Small	[43]	Foodini 3D food printer	Extrusion-based printer (standard)	4.00	//	//	//
		[83]	Custom printer Arduino Mega2560 coupled with a RAMPS 1.4; the software used was Marlin™	Extrusion-based printer (custom)	0.19	//	//	//
		[36]	Delta Wasp 2040	Extrusion-based 3D printer with custom nozzle and extrusion system	6.00	15.00	6.00	3.5 bars
		[44]	Focus 3D food printer	Extrusion-based printer (standard)	1.60	10.00	1.12	//
		[63]	Creality Ender 5 Pro	Extrusion-based printer with Custom Nozzle system for ink deposition	0.60	10 to 20	0.1 to 1	2 to 3 bars
		[45]	Shotmini 200 Sx DIW printer	Extrusion-based printer with 50 mL Luer lock dispensing syringe	0.90	50.00	0.40	0.9 bar
		[46]	3D food printer CARK (controlled additive manufacturing robotic kit)	Extrusion-based printer custom printer for 3D food printing	0.5 to 1.28	5 to 20	//	4 bars
		[47]	Foodini 3D food printer	Extrusion-based printer (standard)	1.50	//	//	//
		[48]	M4 3D printer [87]	Custom printer with seven printheads (two FFF heads, two DIW heads, two IJ heads, and one AJ head), two in situ curing modules (photonic and UV), and two robotic arms	1.194, 0.838, 0.603	//	0.40	//
		[49]	System 60 M	Extrusion-based printer (standard)	0.70	20.00	0.50	//
		[50]	3D printer 3.0 developed by Felix	Extrusion-based printer (standard)	1.55	10	0.60	//
		[38]	KUKA KR 15/2 6-axis industrial robot	Custom extrusion-based printer with a single extruder attached to the robotic arm	5 to 25	20 to 30	//	1.6 to 2 bars
		[80]	3D4E	Extrusion-based custom-made printer capable of printing pastes, gels and highly viscous liquids	0.68	//	//	//
		[58]	Custom LDM printer with syringe-based extrusion system	Extrusion-based custom-made printer	0.61	10.00	0.40	//
		[33]	Foodini 3D food printer	Extrusion-based printer (standard)	8, 15, 40	25 to 58.33	0.7 to 2.8	//
		[57]	Foodini 3D food printer	Extrusion-based printer (standard)	//	33.33	1.95	//
		[52]	3D food printer CARK (controlled additive manufacturing robotic kit)	Extrusion-based printer custom printer for 3D food printing	1.20	1.67 to 10	0.45 to 1.07	2 to 6 bars
		[51]	3D food printer CARK (controlled additive manufacturing robotic kit)	Extrusion-based printer custom printer for 3D food printing	1.20	6.67 to 11.67	0.31 to 1.34	3.2 bars
		[41]	Delta Wasp 2040	Extrusion-based 3D printer with custom nozzle and extrusion system	6.00	15.00	6.00	3.5 bars
		[54]	Shinnove-E Pro	Extrusion-based printer (standard)	0.84	25.00	0.60	//
		[55]	Shinnove-E Pro	Extrusion-based printer (standard)	0.84	25.00	0.60	//
		[56]	Foodbot-D2	Extrusion-based printer (standard)	1.20	20.00	//	//
LDM-Syringe and gear based	Small	[53]	Syringe-based 3D printer 3.0 Felix and gear-based 3D printer L3D Extruder Kit	Extrusion-based printer (standard)	1.55 for syringe-based and 0.5 for gear-based printing	10.00	0.60	//

#### 3.3.2. Selective Laser Sintering AM method

Selective laser sintering is an additive manufacturing method that fabricates three-dimensional objects by successive layers made by laser sintering [60]. In total, 3 of the 57 selected papers were based on SLS and used the AFS-360 rapid prototyping equipment, a commercially available large printer (Table 5). The choice of feedstock materials has a significant impact on the mechanical strength and the surface quality of the sintered or printed object. Currently, SLS feedstock materials are focused on metal, ceramic, polymers, and their corresponding composites. However, the growth of SLS is very limited because of high price and lack of diversity of these feedstock materials. Therefore, there is an urgent need to develop environmentally friendly, cost-effective, and sustainable materials suitable for SLS [30,60,61].

Values of layer height used in SLS printing were between 0.1 to 0.15 mm. Laser power, scanning speed, and scan spacing are important printing parameters for SLS printing. In one of the selected papers, preheating of the powder materials was performed to obtain better print quality [30].

#### 3.3.3. Stereolithography AM Method

Stereolithography (SLA) involves curing a photocurable liquid resin in a tank by the action of an ultraviolet (UV) laser, whose length in commercial systems is typically 405 nm. Stereolithography is different from FFF because it is more accurate and has higher resolution [81]. Table 6 shows that three papers used stereolithography printing. Stereolithography printers are small in size. Irradiation dose, intensity, exposure time, and layer height are important parameters for stereolithography printing. Ethyl cellulose macromonomer (ECM) and rosin-based monomer (DAGMA), photocurable resin, and soyabean oil epoxidized acrylate are used for printing, and the printability of these materials incorporating biowastes was investigated [31,35,81]. The reported studies show that post-processing is required for SLS printing [31].

#### 3.3.4. Binder Jetting, Ink Jetting, and DCW AM Methods

Individual layer fabrication (ILF) involves constructing parts by layering solid materials in a laminated manner. The individual panels are produced separately using a technique called binder jetting. This enables the application of pressure to these panel-like components, similar to the manufacturing process of traditional wood composite boards. Depending on the chosen binder, either hot or cold pressing methods can be employed [88]. The Direct Cryo Writing (DCW) combines 3D printing and cryotemplating, in which ink is extruded onto a cold plate and carefully frozen. All these printing methods, along with the ink jetting method, were used for printing wood powder or chips. Table 7 shows the printers used, printer types, nozzle sizes, and parameters important for different printing methods.

### 3.4. Applications and Limitations

The primary objective of the selected papers was to develop printable filaments, inks, or powders that exhibit mechanical and thermal properties comparable to those of commercially available materials. Therefore, this section emphasizes the sectors in which the incorporation of these biowastes can effectively mitigate the reliance on petrochemical-based materials. Additionally, the section addresses the limitations that currently impede the widespread utilization of biowastesin large-scale applications using additive manufacturing methods.

Approximately 91% of the selected papers in this systematic review focused on material composites that had not been previously utilized (Figure 7a). Conversely, about 9% of the papers explored materials that had been employed in other sectors but had not yet been applied using additive manufacturing methods (Figure 7a). All of these materials have potential applications across various sectors. Selected papers that have not mentioned any potential application sector are mentioned as “Application sector unspecified” in Table 8. Those papers were more focused on developing sustainable filaments, inks, or powders instead of focusing on specific application sectors. Figure 7b presents a breakdown of the sectors. The food industry emerged as the most common area where AM methods could be employed for production utilizing agriculturally derived biowastes. About 38% of the selected papers did not mention specific application sectors, while 23% indicated future applications in the food industry.

According to Table 8, most of the papers that used FDM for printing were focused on developing sustainable, recyclable, and biodegradable 3D printable filaments with biowastes. Some of the papers have shown good results in developing cheaper filaments, lightweight filaments, filaments that can be used in biomedical devices [29], fluorescent emitting 3D printing filaments [76], and filaments that can be customizable and used for producing non-structural components in electrical and automotive industries [65].

Table 8 also shows that the papers using LDM are focused on developing 3D printable ink in food [46,57]. Some of the papers successfully developed printable materials for LDM, which can be used for bone tissue generation [64], food packaging [51], construction sectors [52], waste water treatment plants [83], electrical, pharmaceutical parts production [63], furniture [32,38], and automotive industries [35]. Papers that used SLS were more focused on developing sustainable, low-cost, and environmentally friendly feedstock. Similarly, the papers using the stereolithography method were focused on developing environmentally friendly materials by replacing petrochemical-based materials.

There are several challenges associated with utilizing agriculturally derived biowastes in large-scale industrial settings using various AM methods.

Energy consumption increases when using AM methods compared to other manufacturing methods when the preprocessing (milling, chemical processing, using single or dual screw extruders) and post-processing (polishing, curing, heating, and cooling) steps required for processing agriculturally derived biowastes are included [2,39,60].Particle sizes and fiber sizes of the biowastes can negatively impact the mechanical and thermal properties of the printed products and adversely affect the printing process, i.e., larger fiber degrade elastic modulus, tensile strength [67,69,81,82].Printability window for printable biowastes is narrow in some cases [41,43].An additional notable issue involves the possibility of biowastes undergoing biodegradation or decay over time. Due to the organic components present in biowastes, they can be prone to microbial growth, moisture absorption, and degradation. Consequently, the long-term durability and stability of printed objects may be compromised, thereby constraining their suitability for applications that demand extended lifespans [42].

### 3.5. Mechanical Characterizations 

Figure 8 shows that around 14% of the selected papers did not conduct any mechanical characterizations. About 23% of selected papers used texture profile analysis (TPA). TPA is mainly conducted on food materials so that the yield stress, adhesiveness, hardness, compressibility, and springiness can be measured. But in some cases, TPA has been used to characterize viscoelastic materials for AM methods that might be used in construction, packaging, and furniture industry. Approximately 63% of the selected papers conducted tensile, compression, impact Charpy, flexural, and three-point bending tests. One of the papers showed that in the case of flexural and impact tests, the filler efficiently served as reinforcement; for polylactic acid (PLA)/Mater–Bi^®^ EF51L (MB), the inclusion of anchovy fishbone powder (EE) results in an increase in flexural modulus by roughly 23% and 32%, respectively [42]. Another paper showed that the tensile strength of the 3D printed samples was kept or even improved upon increasing the plant waste content as a biofiller. The elastic modulus was enhanced and only a small reduction in the elongation at break was found [82]. In another paper, white marble powder, eggshell powder, and walnut shell powder were added in PLA. It has shown that the inclusion of eggshell and white marble powder negatively impacted tensile strength. The tensile strength of composites made of biowaste can be increased by mixing walnut shell powder with additional fillers [69]. Another paper showed that the CSP/PLA composites have significantly higher mechanical strength and toughness than pure PLA [71]. Another paper showed that HEMP and WEED compounds improved their performance in terms of mechanical properties and weight reduction compared to conventional thermoplastic materials [29]. According to the data of another paper, nanosized hydroxyapatite (bio-HA) from natural sources is a promising material to utilize in the 3D printing scaffolds with regulated porosity for bone tissue engineering. Pure poly(ε-caprolactone) (PCL) components have a lower modulus (177 MPa), highlighting the stiffening effect of the bio-HA nanoparticles. Moreover, compared to pure PCL, the PCL/bio-HA scaffolds have higher bioactivity. Among these, it was discovered that mussel-shell-derived HA had improved cell adhesion, activation, and proliferation [64].

Effective material characterization is essential for ensuring the quality and performance of 3D printed parts [59,60]. Through a comprehensive understanding of material properties, including tensile strength, elasticity, and thermal behavior, manufacturers can optimize printing parameters and enhance mechanical properties to meet specific requirements [29,34,61,62,72,73]. Furthermore, material characterization aids in addressing potential challenges like warping, shrinkage, or poor layer adhesion, enabling adjustments during the printing process to mitigate these issues [3,41]. Ultimately, meticulous material characterization results in reliable and predictable 3D printed parts using biowastes, offering superior quality, durability, and functionality across diverse applications. 

### 3.6. Future Research Opportunities

Section 3.6.1, Section 3.6.2, Section 3.6.3, Section 3.6.4 and Section 3.6.5 describe future research opportunities for improving the print quality, mechanical and thermal properties.

#### 3.6.1. Rheological Studies

Rheology is a scientific discipline concerned with the study of the flow and deformation of matter and is of utmost importance in 3D printing because it governs the flow of material through the nozzle, its deposition onto the build platform, and its solidification into a finished product [96,97]. A thorough understanding of the viscoelastic properties of the material is required to understand its thixotropic behavior, optimize the printing parameters, and identify potential issues [98].

The ability of some materials to become less viscous over time when subjected to mechanical stress, such as stirring or shaking, is known as thixotropy. A few models are used to explain the viscosity behavior of thixotropic materials, such as the Power law model [99], the Hershel–Bulkley model [100], the Cross model [101], and the Carreau–Yasuda model [102]. Rheology can be used to assess the material’s thixotropic behavior and make sure it is appropriate for 3D printing. For instance, a material with poor flow characteristics might not be appropriate for use in a specific printer or for a specific application [103]. Early detection of these problems (such as viscosity variations, low yield stress, flow instabilities, strain recovery and thixotropy) enables the avoidance of time and money lost on unsuccessful prints.

Some of the selected papers selected for this study did not conduct rheological studies [33,37,59,69,70,71,73]. Rotational and capillary rheometers can both be used to evaluate the validity of the Cox–Merz rule [104]. Two commonly used rheological tests for determining the viscoelastic properties of complex fluids are the incremental deformation test and the oscillatory deformation test. In the incremental deformation test, the fluid is subjected to a constant shear rate, causing it to reach a steady state deformation. The resulting shear stress, τ, is measured, providing information about the steady state viscosity, $\eta_{ss}\left(\dot{\gamma }\right)=\frac{\tau }{\dot{\gamma }}$. In the oscillatory deformation test, the fluid is deformed by applying either shear stress, τ(t), or shear strain, γ(t), induced by harmonic periodic oscillations of frequency ω. This test aims to obtain the transfer function, $\gamma \left(\omega j\right)\overset{G\left(\omega \right)}{\to }\tau \left(\omega j\right)$, which can be expressed as G*(ω) = G′(ω) + G″(ω)j. Here, G′(ω) represents the elastic modulus (storage modulus), and G″(ω) represents the viscous modulus (loss modulus). According to Cox and Merz’s study [105], the steady state viscosity $\eta_{ss}(\dot{\gamma })$ and the complex viscosity defined as $\eta *\left(\omega \right)=\frac{\left|G^{*}\left(\omega \right)\right|}{\omega }$, where $\left|G^{*}\left(\omega \right)\right|=\sqrt{G^{\prime }{\left(\omega \right)}^{2}+G^{\prime \prime }{\left(\omega \right)}^{2}}$ is the magnitude of G*(ω), share a similarity in a sense that $\eta_{ss}\left(\dot{\gamma }\right)\approx \eta *{\left.\left(\omega \right)\right|}_{\omega =\dot{\gamma }}.$ In later research [106], they applied the Cox–Merz rule (CMR) to food materials and observed that the dynamic viscosity (η ∗ (ω)) is often significantly greater than the steady viscosity (η). This finding suggests a nonlinear behavior in the response of biomaterials. Several examples and a modified CMR have been provided in a recent article [107].

Rotational rheometers are suitable for measuring the viscoelastic properties of fluids over a wide range of frequencies, whereas capillary rheometers are useful for measuring the shear viscosity of fluids under high shear rates. The choice of rheometer depends on the particular characteristics of the fluid to be measured and the kind of rheological data needed.

#### 3.6.2. Printing Parameter Optimization

An important step to enhance the mechanical properties of FFF printed parts is the optimization of printing parameters [108,109,110,111]. Layer height, air gaps (both within and between layers), infill density, temperature, and printing speed are the most important printing parameters for optimization [112,113]. However, these parameters must be examined together because of the intricate interplay between them [114]. To this end, various design of experiment (DOE) procedures are available in the open literature, with the Taguchi method being one of the most widely utilized along with factorial design, as well as the Box–Behnken design [115].

The majority of the selected papers focused on developing novel 3D printable materials, and, as a result, none of the papers comprehensively optimized the printing parameters [39,42,61,63,67,68,69,70,71,72,73,74,75,76,77,78,79,82]. To facilitate the large-scale production use of these biowastes, it is imperative that these printing parameters are optimized.

#### 3.6.3. Material Composition Optimization

The surface finish, accuracy, and resolution of the printed object are all affected by the composition of the materials used for printing [116,117,118,119]. Properties including heat conductivity, electrical conductivity, and chemical resistance vary amongst different materials. It is feasible to produce parts with specified qualities to satisfy the requirements of particular applications by optimizing the material composition. The material composition can be optimized to strike a balance between performance and cost. Using less expensive materials, for instance, could result in lower prices but also lesser strength or durability in the printed products [51,57,60,78].

The selected papers analyzed in this review study commonly employed PLA and ABS as matrix materials for FDM printing [34,59,62,73]. For food printing, Xanthan gum, Guar gum, psyllium husk, Na-alginate have been used as matrix materials. Different matrix materials at different compositions were used. Different additives can also be introduced to the materials to reduce production costs, enhance material strength, improve printing accuracy, or produce specialized printed parts for specific applications. For biomass–fungi composites, some substrates, along with Ganoderma fungi, have been tested, but different fungi along with different substrates are yet to be investigated. Some selected papers have shown potential application of biowastes in the biomedical sector (producing scaffolds for bone tissue generation, producing parts with antimicrobial properties) [29,64,66,71]. Different hydrocolloids can be used, too. All these different additives need to be investigated along with different compositions, which provides a vast region of research opportunities and potentially inventing new materials for 3D printing.

#### 3.6.4. Numerical Simulations

Numerical simulations can improve the efficiency of the 3D printing procedure by spotting potential issues in advance. Designers and engineers can choose the best printing settings, such as layer thickness, printing speed, and temperature, by modeling the process [120,121]. Numerical simulations can shed light on the behavior of the material during printing. This knowledge is essential for creating novel materials and figuring out their limitations [122,123,124,125]. The price of 3D printing can be reduced with the aid of numerical simulations. Users can find solutions to reduce material waste, cut down on printing time, and boost printing productivity by simulating the printing, which eventually lowers the cost of production as a whole.

Numerical simulations for 3D printing these new biowastes are still an unexplored region. They enable engineers and new researchers to enhance the quality of the printed product, decrease material waste, boost productivity, and optimize the 3D printing process [126,127].

#### 3.6.5. Life Cycle Analysis

Life cycle analysis (LCA) of these biowastes is missing in all the selected papers except one [70]. That paper showed that palm fiber filament composite used less polymer resin and also reduced the volume of Australian royal palm waste that would otherwise be disposed of in the environment, which could have negative effects on the ecosystem. More LCA studies are required for these novel materials so that the environmental impact of these materials can be assessed.

The agriculturally derived biowastes have a great potential for industrial applications. Furthermore, because European standards call for 20% biobased materials, the secondary thermoplastic industry sectors can effectively utilize these new agricultural derived biowastes. Biodegradability, sustainability, environmental and social impacts of these materials need to be investigated and AM methods need to be developed more for industrial sectors. This research can effectively provide a path towards bio-circular economy.

## 4. Conclusions

Papers addressing novel materials made from agriculturally derived biowastes were gathered using a systematic process. The primary goal of this systematic review was to assess the current state of the art with regard to issues related to the particular emphasis on the novel 3D printable materials derived from biowastes, the specifics of 3D printing, and potential applications. Around 58% of the selected papers are from the materials research area.

The majority of biowastes and by-products are derived from agricultural, fishery, forestry, and agrifood industries. A limited number of novel materials have been developed for additive manufacturing thus far. However, researchers are actively exploring new sources of biowaste. A few feedstock materials have been investigated for producing printable filaments, inks, and powders. A large number of selected papers have studied the particle sizes of biowaste for achieving desired printability. Moreover, the size and shape of these particles can affect nozzle clogging, surface roughness, and the presence of defects in the printed parts. PLA was the most used matrix for biowaste-reinforced composites. Guar gum, Xanthan gum, psyllium husk, and other gels were used for creating 3D printable inks. However, this integration poses challenges due to the alteration of material properties, potentially impacting mechanical and thermal characteristics of the printed products. These challenges include reduced strength, increased brittleness, changes in dimensional stability, impaired layer adhesion, interlayer bonding issues, and decreased printability, accuracy, and resolution of complex geometries.

Mechanical and rheological characterizations were conducted in most of the selected papers. Comparable results were found in some cases, whereas favorable results were not obtained in many cases. Material characterization is vital for ensuring the quality and performance of 3D printed parts. By understanding material properties and behavior, manufacturers can optimize printing parameters and enhance mechanical properties. Thorough material characterization enables the identification and mitigation of potential issues, resulting in reliable and high-quality 3D printed parts for various applications.

The majority of selected papers (91%) focused on novel material composites. These materials show potential for various sectors, with the food industry being the most common areas for AM production utilizing agriculturally derived biowastes. Here, 38% of the selected papers did not mention specific application sectors, while 23% indicated future applications in the food industry. The papers using FDM primarily focused on developing sustainable and biodegradable filaments, while the papers using LDM aimed to develop printable ink for food, construction, packaging, and biomedical sectors. Papers using SLS and stereolithography focused on developing sustainable feedstock. 

There are limitations to promote the use of agriculturally derived biowastes and different additive manufacturing methods in large-scale settings. These include higher energy consumption due to pre-processing and post-processing steps, the time-intensive nature of most AM methods, higher material costs compared with other manufacturing methods, variability in biowastes impacting product properties and the printing process, a narrow printability window for many biowastes. Resolving these limitations is crucial for a wider adoption of agriculturally derived biowastes and AM in large-scale applications.

Many of the selected papers mentioned few potential markets for these novel materials. Using biowastes through additive manufacturing has a wide range of applications. The purpose of this review is to encourage more research to utilize agriculturally derived biowastes in the actual world using AM.

## Figures and Tables

**Figure 1 bioengineering-10-00845-f001:**
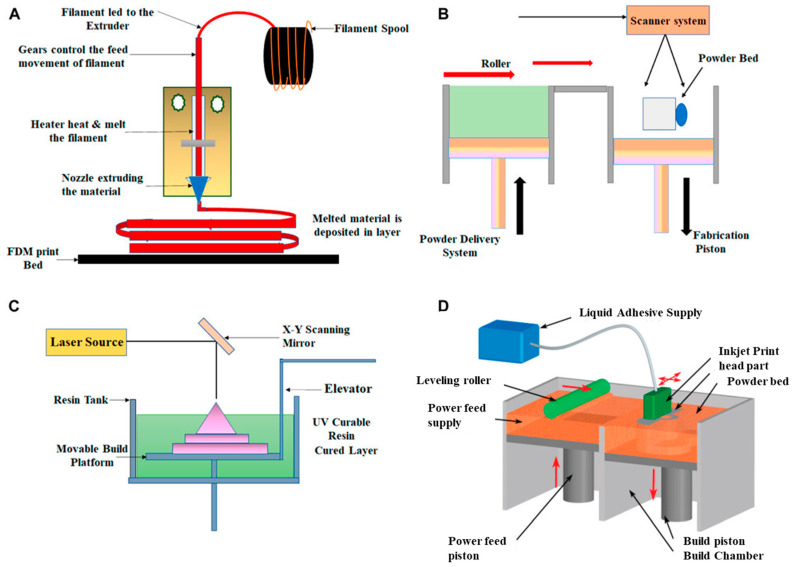
Illustrations of (**A**) fused deposition modeling, (**B**) selective laser sintering, (**C**) stereolithography, and (**D**) binder jetting (redrawn from [23,24]).

**Figure 2 bioengineering-10-00845-f002:**
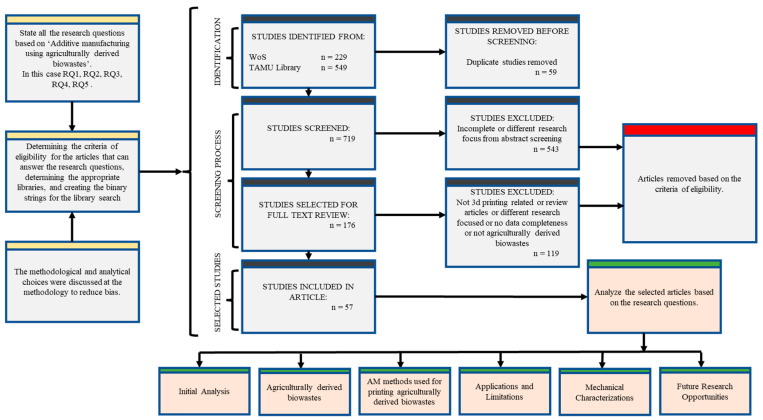
Flow chart of the systematic review.

**Figure 3 bioengineering-10-00845-f003:**
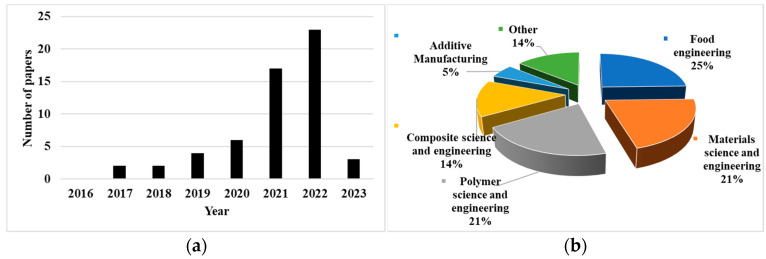
Initial analysis results: (**a**) number of papers in each year, (**b**) percentage of selected papers in each research area.

**Figure 4 bioengineering-10-00845-f004:**
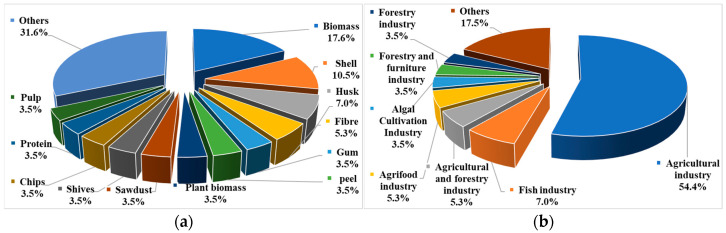
Analysis results showing percentage of selected papers based on (**a**) types of the agriculturally derived biowastes and (**b**) sources of agriculturally derived biowastes.

**Figure 5 bioengineering-10-00845-f005:**
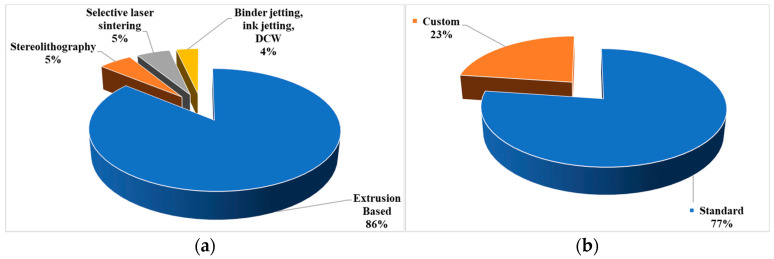
Analysis results showing percentage of selected papers based on (**a**) different AM methods and (**b**) standard or custom 3D printers.

**Figure 6 bioengineering-10-00845-f006:**
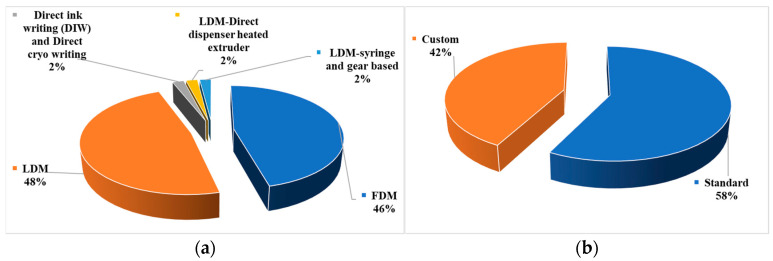
Analysis results showing percentages of selected papers based on (**a**) different types of extrusion-based AM methods and (**b**) standard or custom 3D printers for LDM.

**Figure 7 bioengineering-10-00845-f007:**
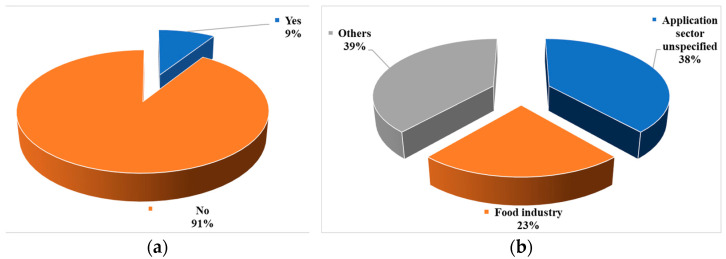
Analysis results showing the percentages of reported papers: (**a**) whether the materials are already used or not, and (**b**) application sectors of biowaste-incorporated materials.

**Figure 8 bioengineering-10-00845-f008:**
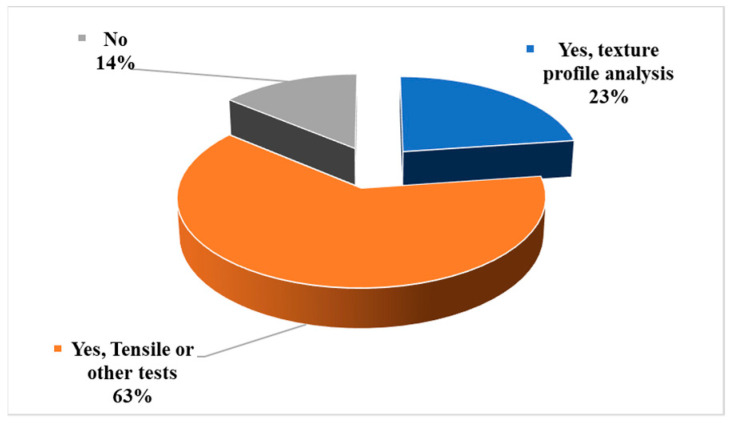
Analysis results showing the percentages of reported papers: whether mechanical characterizations were conducted or not.

**Table 1 bioengineering-10-00845-t001:** Criteria of eligibility, search library and binary strings for query for selecting papers to be included.

Criteria for Eligibility	Search Library	Binary Strings for Query
Have to be accessible from the libraries	TAMU Library	(“Additive manufacturing*” OR “3D print*” OR “am methods*” OR “rapid prototyping*” OR “extrusion-based method*” OR “3d-print*” OR “3D printing” OR “3D-printing” (All Fields) AND “agriculturally derived materials*” OR “agri-food*” OR “agro-food*” OR “agrofood*” OR “agroindustrial*” OR “food*” OR “agricult*” (All Fields) AND “feedstock” OR “biomass*” OR “bio-mass*” OR “biowaste*” OR “waste*” OR “biomass-fungi*” OR “biomass-fungi composites*” OR “biocomposites*” OR “scrap” OR “biomass fungi*” (All Fields))
Have to involve AM methods for manufacturing products		
Have to involve agriculturally derived biowastes Papers cannot be review papers		
Have to focus on experiments, characterizations, or applications	Web of Science (WoS)	(((ALL = (“Additive manufacturing*” OR “3D print*” OR “am methods*” OR “rapid prototyping*” OR “3d-print*” OR “3D printing” OR “3D-printing”)) AND (ALL = (“feedstock” OR “biowaste*” OR “biomass*” OR “bio-mass*” OR “waste*” OR “biomass-fungi*” OR “biomass-fungi composites*” OR “biocomposites*” OR “scrap” OR “biomass fungi*”)))
Have to be in English		

**Table 2 bioengineering-10-00845-t002:** A list of selected papers organized by research area and year of publication.

Principal Research Area	2016	2017	2018	2019	2020	2021	2022	2023
Additive Manufacturing	//	//	//	//	[29]	[30]	[31]	//
Agriculture and food science	//	//	[32]	//	//	//	//	//
Biological science	//	//	//	//	//	//	[33]	//
Biomaterials engineering	//	//	//	//	[34]	//	//	//
Chemical engineering	//	//	//	//	[35]	//	//	//
Composite science and engineering	//	//	//	//	[36]	[4,37]	[38,39,40,41,42]	//
Food engineering	//	//	//	[43]	[44]	[45,46,47]	[48,49,50,51,52,53,54]	[55,56]
Food Science and Nutrition	//	//	//	//	//	//	[57]	//
Green chemistry and engineering	//	//	//	//	//	//	[58]	//
Materials science and engineering	//	[59,60]	[61]	[62]	//	[63,64,65,66]	[67,68,69,70,71]	//
Polymer science and engineering	//	//	//	[72]	[73]	[3,74,75,76,77,78,79]	[80,81]	[82]
Water research	//	//	//	[83]		//	//	//

**Table 3 bioengineering-10-00845-t003:** List of selected papers by Biowaste Types, Biowaste Sources, Biowaste, Matrix, and Weight Percentages.

Biowaste Types	References	Biowaste Sources	Biowastes	Matrix	% w.t.
Anthocyanin antioxidant	[47]	Purple sweet potato (PPP)	Purple sweet potato powder, mulberry powder, carrot powder, black wolfberry powder, roselle powder	Pulped yam	0.5 to 6
Bean	[53]	*Phaseolus vulgaris* L.	Protein extract from common bean	Sodium alginate, gelatin, water for syringe-based 3D printing; agar, xanthan, water for gear-based 3D printing	40 for syringe based and 10 for gear based
Biomass	[32]	Algae	Nostoc sphaeroides	Water/Juice	5
	[36]	Switch grass, rice straw, sorghum stalks, and hemp	Biomass	Water	40
	[35]	Wood and rosin	Ethyl cellulose macromonomer (ECM) and rosin-based monomer (DAGMA)	HEA/DAGMA	//
	[44]	*Arthrospira platensis*	Antioxidants	Wheat flour, butter, powdered sugar, milk, xanthan gum	0.8 to 10
	[65]	Miscanthus	Biocarbon	Poly(trimethylene terephthalate)	5 to 10
	[77]	Oil palm empty fruit bunch	Organosolv lignin	Acrylonitrile butadiene styrene (ABS)	5 to 15
	[80]	Corn	Hemicellulose and lignin	Water	76 to 86
	[33]	Spirulina (arthrospira platensis) and/or chlorella vulgaris	Chlorella vulgaris and arthrospira platensis (“spirulina”) biomass	Corn, rice flours, olive oil and water	5 to 30
	[40]	Corn	Lignocellulosic corncob	Polyhydroxybutyrate (PHB)/polylactic acid (PLA) biopolymer	0 to 8
	[41]	Switch grass, rice straw, sorghum stalks, and hemp	Biomass	Water	40
Bone and shell	[64]	Cuttlefish, egg, mussel	Nanometric hydroxyapatite (HA)	Poly(ε-caprolactone) (PCL)	15
Bran	[67]	Wheat	Wheat wastes (middlings of bran)	Poly lactic acid (PLA)	10
By products	[43]	Cod	Surimi	Water	25
Chips, stalks	[74]	Wood	Wood powder	Wood powder and adhesives	//
	[39]	Lignin from industrial waste	Ethyl acetate treated lignin nanospheres (EALNSs)	Poly lactic acid (PLA)	0.50
Fiber	[70]	Royal palm	Palm fiber	Acrylonitrile butadiene styrene (ABS)	5 to 20
	[72]	Skin of flax plant	Flax fiber	Poly lactic acid/polybutylene adipate terephthalate	10 to 30
	[73]	Vegetable	Hydroxypropyl methylcellulose (HPMC)	Poly lactic acid (PLA)	1 to 7
	[75]	Ramie plants	Ramie fiber	Poly lactic acid (PLA)	//
Fishbone	[42]	*Engraulis encrasicolus* (EE) fish	Anchovy fishbone powder	Polylactic acid (PLA)/mater-bi^®^ ef51l (mb)	10 to 20
Flour	[62]	Wood	Wood flour particles	Different printing methods had different matrix	//
Gum	[55]	Wheat, corn, soy and dairy	Xanthan and guar gum	Soy protein isolate emulsion gel	0.2 and 0.5
	[56]	Wheat, corn, soy and dairy	Konjac gum (KGM)/xanthan gum (XG)	Water	0.15 to 0.9
Hulls	[4]	Soy	Biocarbon	Recycled high-density polyethylene (HDPE) and polypropylene (PP)	20
Husk	[37]	Buckwheat	Buckwheat husk	Poly lactic acid/polybutylene adipate terephthalate	5 to 15
	[30]	Peanut	Peanut husk powder	Polyether sulfone	10 to 25
	[3]	Rice	Rice husk fiber	Recycled polypropylene	5 to 10
	[58]	Corn	Corn starch and cellulose fiber	Water	34 to 44
Oil	[31]	Vegetable	Soybean oil epoxidized acrylate	Soyabean oil epoxidized acrylate	//
Peel and bagasse	[46]	Potato	Potato peel powder	Guar gum, whole wheat, table salt, vegetable oil	0 to 100
	[52]	Banana	Banana peel	Banana peel paste	40
	[51]	Banana and sugarcane	Banana peel (BP) and sugarcane bagasse (SCB)	Banana peel and sugarcane bagasse paste	10 to 90
Plant biomass	[79]	Opuntia Ficus indica	Cladodes	Polylactic acid (PLA)	//
	[82]	Solanum *Lycopersicon* plant	Lignocellulosic wastes	Mater-Bi^®^ EF51L (MB)	5 to 15
ProteinIsolate	[50]	Soy	Soy protein isolate	Water and Na alginate solution	20
	[54]	Soy	Soy protein isolate (SPI)	Water	6
PulpBiomass	[48]	Wood	Cellulose nanocrystals	Tomato, spinach, and applesauce puree	2.5 to 7.5
	[63]	Wood, cotton, hemp	Ethyl cellulose	A-terpineol	8
Sawdust	[34]	Wood	Wood powder	Polylactic acid (PLA)	30
	[38]	Beechwood	Biomass	Water	5–28.5
Shell	[59]	Macadamia nut	Micro-ground macadamia nutshell polymer composite	Acrylonitrile butadiene styrene (ABS)	//
	[60]	Walnut	Walnut shell powder	Copolyester hot melt adhesive (co-pes)	0 to 52
	[61]	Walnut	Walnut shell powder	Copolyester (co-pes) powder, copolyamide (co-pa)	40
	[83]	Crabs and other crustaceans	Chitosan	Water and glacial acetic acid	2
	[69]	Wall nut and egg	Powder from eggshell, walnut shell, and white marble	PLA and abs with different biofillers	2.5 to 5
	[71]	Crabs and other crustaceans	Crab shell powder	Poly (lactic acid) (PLA)	1.50
	[78]	Cocoa bean	Cocoa bean shells	Recycled polypropylene	5.00
Shives	[29]	Industrial plant	Weed, hemp	Poly lactic acid	Hemp 15–25, Weed 10 to 15
	[76]	Flax	Fluorescent rafted flax shives (FG-FS) and flax shives (FS)	Poly-(butylene-terephthalate) (PBAT)	10
Skin	[49]	Seafood	Gelatin	Water	2 to 14
Soybean byproduct	[45]	Soybean	Okara	Water	25 to 50
Stalk	[68]	Kenaf	Kenaf straw core (KSC) and kenaf fiber (KF)	Fly ash (FA), ground granulated blast furnace slag (GGBFS) (geopolymer)	Ksc 1.5 and kf 0.2
Starch	[57]	Potato, corn, vegetables	Carbohydrate	Beef	//
Strain	[66]	Polymorphic fungus *Aureobasidium pullulans*	Pullulan (PUL)	Poly(3-hydroxybutyrate-co-hydroxy valerate) (PHBV), hydroxyvalerate (HV)	5
Straw	[81]	Rice, wheat	Straw fiber	Photocurable resin	5

**Table 5 bioengineering-10-00845-t005:** List of the selected papers by 3D Printer Name, 3D Printer Type, Layer Height, Specifications for SLS.

Reference	3D Printer Name	Layer Height (mm)	Specifications for SLS
[60]	AFS-360 rapid prototyping equipment	0.10	Laser power of 14 W, scanning speed of 2000 mm/s, layer thickness of 0.1 mm, scan spacing of 0.2 mm
[61]	AFS-360 rapid prototyping equipment	0.15	Wavelength of 10.6 μm and laser power of 55 W, Scan speed 2000 mm/s, scan spacing 0.2, laser power 12
[30]	AFS-360 rapid prototyping machine	0.15	Laser wavelength 10.6 micrometer, scan speed 2000 mm/s, scan spacing 0.2 mm, processing temperature 75, preheating 82 and laser power 14 W

**Table 6 bioengineering-10-00845-t006:** List of the selected papers by 3D Printer Name, 3D Printer Type, Specifications for Stereolithography Printing.

References	3D Printer Name	Specs for Stereolithography Printing
[35]	Creality LD 001	//
[31]	Original PRUSA SLI	Irradiation dose 0.75–1.5 mj·cm^2^, intensity 0.1 mw/cm^2^ and exposure time 7.5–35 s per 0.05 mm layer
[81]	Forms Lab 1	405 nm laser

**Table 7 bioengineering-10-00845-t007:** List of the selected papers by AM Methods, Printer Size, 3D Printer Name, 3D Printer Type, Nozzle Sizes for DIW and DCW (mm), Specifications for Ink Jet Printing, and Specifications for Binder Jet Printing.

AM Methods	References	Printer Size	3D Printer Name	3D Printer Type	Nozzle Sizes for DIW and DCW (mm)	Specifications for Ink Jet Printing	Specifications for Binder Jet Printing
Binder jetting, Direct ink writing (DIW) and Direct cryo writing (DCW)	[62]	Large, small	Hyrel3D 30M, Dimatix DMP-2831 piezoelectric ink jet printer, Cometrue T10 binder jet printer	Extrusion-based printer with custom nozzles and inkjet printer with a 10 pL cartridge which had 16 nozzles along with a binder jet printer	For DIW and DCW, the nozzle was 1.2 mm	Ink jetting frequency was set to 1000 Hz using 200 or 600 dpi. Platform temperature was set to 60 °C.	High-resolution printing setup was selected, with level 3 counter width and layer height of 0.08 mm for binder jetting.
Individual Layer Fabrication (ILF)	[74]	Large	Custom modified for Individual Layer Fabrication	Binder jetting-based system	//	//	Brush roller and a scatter roller was used. Electro-pneumatically driven jet valve system was used for adhesive distribution.

**Table 8 bioengineering-10-00845-t008:** List of the selected papers by AM Methods, Application Sectors, New Materials, and Product Applications.

AM Methods	Application Sectors	References	New Materials	Product Applications
Binder jetting, Direct ink writing (DIW) and Direct cryo writing	Construction sector for thermal insulation	[62]	Yes	Developing 3D printing materials replacing synthetic binder and using 100% wood extracts
Binder jetting—ILF	Construction industry	[74]	No, but new in 3D printing process.	Wood panels
FDM	Biomedical industry	[29]	Yes	Biomedical devices
		[71]	Yes	Developing sustainable bio-based Dd printable filaments
		[66]	Yes	Developing 3D printing filament using pullulan to apply in tissue engineering
	Electrical and automotive industry	[65]	Yes, never used BC in PTT matrix	Customizable, non-structural components in electrical and automotive industries
	Application sector unspecified	[59]	Yes	3D printing filament improvement for lightweight print
		[72]	No, already used in [89]	Biocomposite production
		[73]	Yes	3D printing filament development using biofillers
		[34]	No	Developing sustainable bio-based 3D printable materials (potential exterior use)
		[77]	Yes, graphene fillers added with biofiller	Improving 3D printing materials interlayer adhesion properties
		[75]	No, already used in [90] [91]	Biodegradable filament production
		[37]	Only shown as representative example of a wide range of lignocellulosic waste that could be used as an alternative filler	Cheaper filament production
		[76]	Yes	Fluorescent emitting 3D printing filament development
		[79]	Yes	Developing sustainable bio-based 3D printable materials for green fabrication of furniture panels, objects, toys
		[4]	Yes	Developing filaments using 100% recyclable plastics and soybean residues
		[78]	Yes	3D printable composite filaments based on agro-industrial and polymeric wastes such as cocoa bean shell (CBS)
		[3]	Yes	3D printable composite filaments based on agro-industrial and polymeric wastes such as rice husk
		[70]	Yes	Developing cheaper 3D printable filaments
		[42]	Yes	3D printable composite filaments based on fish residue such as fish bone powder
		[69]	No	Developing biofilled filaments for 3D printing
		[67]	No	3D printing filament development using biofillers
		[39]	Yes	Improving filaments for 3D printing
		[40]	Yes	3D printable composite filaments based on agro-industrial and polymeric wastes such as corncob biomass
		[82]	Yes	Developing sustainable bio-based 3D printable materials
LDM	Biomedical industry	[64]	No	Bone tissue generation
	Construction sector	[68]	Yes	Developing crack resistance, high shape retention and low carbon emission of 3D-printed material
	Construction, furniture industries	[38]	No	Biomass–fungi biocomposite 3D printing
	Food industry	[32]	Yes, but already used for traditional food in different countries	3D food printing for controlled nutrition supply
		[43]	Yes	3D fish printing
		[44]	No	Developing 3D printable food ink
		[46]	Yes	Food printing
		[47]	Yes	Developing color 3D printable food ink
		[45]	No	Developing okara ink as food without additive to alter rheological properties
		[57]	No, but new for beef 3D printing	Using oxidized starch to improve 3D food printing quality
		[33]	No	Gluten-free snack production
		[48]	Yes, because previously used to create biocompatible cell culture scaffolds, high-strength aerogel structures, and packaging	3D food printing for controlled nutrition supply
		[49]	Yes	Food printing
		[54]	No	Developing 3D-printable food ink
		[50]	No	Protein-based food production
		[55]	No	Developing 3D-printable food ink
		[56]	Yes	Developing 3D-printable food ink
	Food, electronic, pharmaceutical industry	[63]	Tested in different papers [92,93]	Hypothesis for food printing, flexible electronic parts, tablet printing
	Application sector unspecified	[36]	Yes	Developing printable materials for packaging, construction, furniture
		[80]	No, already used in [94,95]	3D printing crude lignocellulosic biomass extracts
		[58]	Yes	Developing thermally tunable sustainable 3D printing inks
	Packaging industry	[41]	No	Packaging, construction, furniture industry
		[51]	Yes	Developing 3D printable food packaging materials
		[52]	Yes	Developing 3D printable food packaging materials
	Wastewater treatment plants	[83]	Yes	Hypothesis to use in wastewater cleaning sector for cleaning pollutants such as amoxicillin
Selective laser sintering	Biomedical, construction, electronic and aerospace industry	[30]	Yes	Producing environmentally friendly SLS materials to use in manufacturing medical equipment, automotive parts
	Application sector unspecified	[60]	Yes	Developing sustainable, low-cost, and environmentally friendly feedstock for SLS printing
		[61]	Yes	Developing sustainable, low-cost, and environmentally friendly feedstock for SLS printing
Stereolithography	Aerospace, automotive, and electronics industries	[35]	Yes	Hypothesis to use in flexible conductive hydrogels that have important potential application in the flexible electronic materials and smart photoelectric materials and developing sustainable and green polymeric 3D printable materials for replacing petroleum-based materials use
	Construction sector	[81]	Yes	Developing composites for stereolithography which can be used for thermal insulation
	Application sector unspecified	[31]	Yes, because nanocellulose has been used as a filler in the resin	Reducing petrochemical use and new resin production
LDM—Syringe and gear based	Food industry	[53]	Yes	Developing 3D printable food ink

## Data Availability

The authors confirm that the data to support the findings of this study are available within the article or upon request to the corresponding author.

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
