# Peer review of "Additive Manufacturing Using Agriculturally Derived Biowastes: A Systematic Literature Review"

_bioengineering, 2023, doi:10.3390/bioengineering10070845_

Round 1
Reviewer 1 Report
1. The manuscript lacks a clear research objective and fails to establish a strong rationale for studying the development of novel materials from agriculturally derived biowastes for additive manufacturing. See this and discus it: Biopolymeric sustainable materials and their emerging applications, Journal of Environmental Chemical Engineering
2. The literature review presented in the manuscript is insufficient and does not adequately cover recent and relevant studies in the field of agriculturally derived biowastes for additive manufacturing.
3. The methodology used to select the 57 publications for analysis is unclear, leading to potential bias and a limited representation of the current state of the field.
4. The manuscript does not provide a comprehensive analysis of the material properties, processing techniques, and end-use applications of the transformed biowastes. It lacks critical evaluation and fails to address the limitations and challenges associated with utilizing these materials.
5. The manuscript lacks originality and fails to contribute significantly to the existing body of knowledge on agriculturally derived biowastes in additive manufacturing. The findings presented do not offer novel insights or advancements in the field.
6. The discussion of potential applications for the transformed biowastes is superficial and lacks practical relevance. The manuscript does not provide a clear understanding of how these materials can be effectively integrated into real-world additive manufacturing processes.
7. The manuscript fails to address the environmental impact and sustainability aspects associated with utilizing agriculturally derived biowastes in additive manufacturing. The authors should consider the broader implications of their research and provide a more comprehensive analysis.
8. The writing style of the manuscript is unclear, with several grammatical and structural errors. The lack of clarity and coherence hinders the reader's understanding of the research objectives, methodology, and findings.
9. The manuscript lacks strong conclusions and recommendations for future research. The authors should clearly outline the implications of their findings and suggest specific areas for further investigation to advance the field.
10. Overall, the manuscript does not meet the scholarly standards required for publication. It lacks rigor, originality, and a comprehensive analysis of the topic. Significant revisions and improvements are necessary to make it suitable for publication in a reputable journal.
Extensive editing of English language required
Author Response
A major revision has been done by the authors based the reviewer's comments.

Reviewer 2 Report
The manuscript is a literature review of recent research works on additive manufacturing processes of agriculturally derived biowastes.
The topic discussed in this review is very interesting. The language of the manuscript is good. Results presented in this manuscript are of great interest.
However, in some sections authors present a gather of literature works without discussion of them. So, I suggest that authors reinforce the discussion aspect because it is essential in a review article.
This paper could be accepted for publication in bioengineering after authors address the above given issues:
* Detailed comments are reported on the attached file : Reviewcomments.pdf

The language of the manuscript is ood and clear.
Author Response
A major revision has been done by the authors based on the reviewer's comments.

Reviewer 3 Report
The presented review work emphasized the utilization of bio wastes into useful products through additive manufacturing. The contents and technical aspects of the work is really interesting and extremely important for achieving sustainability goals to minimize the waste materials. However, following minor corrects are required to improve the quality of the manuscript. I request mandatory revision, as listed below, please do not simply respond but revise manuscript.
· An abbreviation section may be included at the beginning of the Introduction and it should indicate all the necessary terms used in this work. It may help the readers to easily understand the important abbreviations used in this work.
· A detailed flowchart (It may include the main subdivisions of the reviewed work of this manuscript) may be included in the Methodology section to describe the flow of the proposed work. It may help the readers to understand how the authors have distinguished their review work.
· Conclusion is looks like comprehensive discussion. It should be shortened with important findings of the work and future scopes.
Author Response
Revision has been done by the authors based on the reviewer's comments.

Round 2
Reviewer 1 Report
Accept in revised form
Accept in revised form